

# Effect of snow microstructure variability on Ku-band radar snow water equivalent retrievals

Nick Rutter[1], Melody J. Sandells[2], Chris Derksen[3], Joshua King[3], Peter Toose[3], Leanne Wake[1], Tom Watts[1], Richard Essery[4], Alexandre Roy[5], Alain Royer[6], Philip Marsh[7], Chris Larsen[8], Matthew Sturm[8]

[1]Department of Geography and Environmental Sciences, Northumbria University, Newcastle upon Tyne, UK.
[2]CORES Science and Engineering Limited, Burnopfield, UK.
[3]Climate Research Division, Environment and Climate Change Canada, Toronto, Canada.
[4]School of GeoSciences, University of Edinburgh, UK.
[5]Département des Sciences de l'Environment, Université du Québec à Trois-Rivières, Canada
[6]Département de Géomatique Appliquée, Université de Sherbrooke, Canada.
[7]Department of Geography, Wilfrid Laurier University, Canada.
[8]Geophysical Institute, University of Alaska, Fairbanks, USA.

*Correspondence to*: Nick Rutter (nick.rutter@northumbria.ac.uk)

## Abstract

Spatial variability in snowpack properties negatively impacts our capacity to make direct measurements of snow water equivalent (SWE) using satellites. A comprehensive data set of snow microstructure (94 profiles at 36 sites) and snow layer thickness (9000 vertical profiles across 9 trenches) collected over two winters at Trail Valley Creek, NWT, Canada, were applied in synthetic radiative transfer experiments. This allowed robust assessment of the impact of first guess information of snow microstructural characteristics on the viability of SWE retrievals. Depth hoar layer thickness varied over the shortest horizontal distances, controlled by subnivean vegetation and topography, while variability of total snowpack thickness approximated that of wind slab layers. Mean horizontal correlation lengths were sub-metre for all layers. Depth hoar was consistently ~30% of total depth, and with increasing total depth the proportion of wind slab increased at the expense of the decreasing surface snow layer. Distinct differences were evident between distributions of layer properties; a single median value represented density and SSA of each layer well. Spatial variability in microstructure of depth hoar layers dominated SWE retrieval errors. A depth hoar SSA estimate of around 7% under the median value was needed to accurately retrieve SWE. In shallow snowpacks <0.6m, depth hoar SSA estimates of ±5-10% around the optimal retrieval SSA allowed SWE retrievals within a tolerance of ±30 mm. Where snowpacks were deeper than ~30cm, accurate values of representative SSA for depth hoar became critical as retrieval errors were exceeded if the median depth hoar SSA was applied.



## 1 Introduction

Seasonally snow covered, non-glaciated, Arctic terrestrial environments north of tree line cover approximately 5.05 x $10^6$ km$^2$ (Walker et al., 2005). Layering of snow, where distinct differences in snow properties exists between vertically adjacent strata (Fierz et al., 2009), is spatially heterogeneous in Arctic regions with a dense wind slab layer overlaying less dense depth hoar

(Benson and Sturm, 1993- see Fig.1; Derksen et al., 2009). As depth hoar and wind slab have strongly diverging microwave scattering properties (Hall et al., 1991), the relative proportion of each strongly influences Ku-band radar backscatter (Yueh et al., 2009; King et al., 2015; King et al., 2018). Knowledge of how layers vary within Arctic snowpacks is therefore critical to the assessment of uncertainty in radar-based retrievals of snow water equivalent and forward models of snow radiative transfer. Subnivean topography, wind redistribution and vertical thermal gradients dominate the formation of layers in Arctic snowpacks

(Benson and Sturm, 1993; Sturm and Benson, 2004). Grassy tussocks are common in tundra environments, allowing early winter snowfall to collect in wind-protected hollows between tussock mounds. Strong thermal gradients in shallow early season snowpacks cause extreme thermal metamorphism required to initiate growth of large depth hoar crystals and chains (Sturm et al., 1997). Growth of depth hoar in this lowest layer of the snowpack then continues throughout the winter, usually more so than in any other snowpack layer as it is subject to strong thermal gradients for the longest period of time. Subsequent snowfall

events throughout the winter contribute to the development of high density layers of wind-compacted snow crystals through aeolian-driven redistribution (Derksen et al., 2014). These wind-compacted layers become prominent when the total snow depth exceeds the height of the tussocks and the snow surface is fully exposed to winds. Tundra shrubs have a similar influence on snow catchment and metamorphism; reducing local wind velocities, providing shelter for early season snow deposition which favours development of depth hoar (Sturm et al., 2001).

Highly heterogeneous microstructural properties of snowpacks are problematic for retrieval of SWE via remote sensing. While passive microwave remote sensing historically has provided estimates of global SWE distributions (Kelly, 2009) much uncertainty exists, largely due to the impact of seasonal snow microstructural evolution on microwave scattering in SWE retrieval models (e.g. Derksen et al., 2014). However, recent experimental work using Ku-band radar suggests using two frequencies, each with different sensitivities to wind slab and depth hoar, may mitigate this primary source of uncertainty

(King et al., 2018; Lemmetyinen et al., 2018). In order to apply this two-frequency approach in a distributed manner, we need an understanding of layer length scales; horizontal distances over which the physical properties of each layer decouple and become statistically uncorrelated to each other. The spatial scales of interest depend on the application. For SWE distribution within tundra catchments, it is critical to understand layer variability at the landscape scale (10-1000 m), where different topographic units (e.g. plateau, slope and valley) are subject to different snowdrift, scour and sublimation processes.

Understanding this resolution of landscape scale variability in snow properties (layers, density and microstructure) is particularly relevant to future active microwave satellite mission concepts, as well as distributed hydrological modelling which use meteorological inputs from high-resolution numerical weather prediction models. While the range of spatial variability in snowpack layering and snowpack properties can be estimated from multiple one-dimensional profiles distributed throughout





a catchment, this alone will not adequately describe variability in horizontal correlation length scales of snowpack properties. Knowing how these correlation length scales change between different landscape topographic units can help upscale centimetre scale field measurements of snow properties characterising spatial uncertainty in the statistical distributions of parameters with strongly different microwave scattering capabilities. Such statistical distributions could then be used in a radiative transfer

model, such as the recently developed snow microwave radiative transfer model (SMRT) (Picard et al., 2018), to address how accurately the information on snowpack properties needs to be known to inform a viable SWE retrieval. Consequently, for the first time, this opens up the potential to explore how variability in snow layers might impact radar backscatter and retrievals of SWE from backscatter; to do so this study will:

    1. Quantify the spatial variability of snow depth and layer thickness for surface snow, wind slab and depth hoar within

the Trail Valley Creek catchment.

    2. Determine representative snow microstructural parameters (SSA, density) and their associated variability for individual tundra snowpack layers.

    3. Use these relationships to construct a series of synthetic snowpacks with a realistic range of parameters to quantify the impact of spatial uncertainty in snow microstructural parameters on SWE retrieval accuracy from radar backscatter

at two frequencies (13.4 and 17.2 GHz) with SMRT.

## 2 Methods

### 2.1 Field data

Field data presented in this study were collected between 4-9 April 2013 and 14-22 March 2018 within the research basin of Trail Valley Creek (TVC), NWT, Canada (68°44'N, 133°33'W) located at the southern edge of the Arctic tundra.

Measurements of snow microstructure were made mainly on graminoid tundra, which dominates the land cover, as well as in patches of taller shrubs (willow or alder) found on south facing slopes and in proximity to drainage channels and water features (Marsh et al., 2010). Snow pit and snow trench locations (Figure 1) were focussed on gently undulating (<5° slope angle) upper tundra areas that were exposed to high winds, while also incorporating some slope and valley bottom areas, representative of more sheltered areas in the catchment. Hourly air temperatures were measured throughout both winters to

provide temporal context of freeze-up, mid-winter melt events and snow melt onset.

### 2.1.1 Snowpack properties

To investigate the spatial variability of snowpack layering within the snowpack at TVC, nine snow trenches each ranging in length from 5 m to 50 m were excavated in 2013. Of the 9 snow trenches, one 50 m (trench 4) and six 5 m trenches were located in gently undulating upper tundra plateau areas, while two 5 m trenches (trench 6 and 7) were located within valley

bottoms (Figure 1). Following the methods of Tape et al. (2010), at each trench, near infrared (NIR) photography was used to quantify two-dimensional changes in snow snowpack layering. The positions of all layer boundaries were subsequently



geolocated at 1 cm resolution throughout the length of each trench (Watts, 2015); a 5m trench therefore provides 500 vertical profiles of snowpack layering. Within each 5 m trench section (including ten 5 m sections of the 50 m trench), measurements of density and specific surface area (SSA) were made in a single vertical profile and subsequently assigned to individual layers, which were assessed through visual inspection and hardness. SSA was measured using both an InfraRed Integrating Sphere

(IRIS) (Montpetit et al., 2012) for the trenches and pits in 2013 and 2018, and an A2Photonics IceCube measurement system for the pits in 2018; both measurement systems followed principles presented in Gallet et al. (2009). Layers were often discontinuous due to the spatial variability of subnivean topography and aeolian processes. One-dimensional vertical profiles of snow properties, in combination with inspection of two-dimensional NIR imagery, allowed individual layers to be manually classified into one of three microstructure types: surface snow (SS), wind slab (WS), or depth hoar (DH) (Figure 2). While the

surface snow layer is often of low density (<100 kg m$^{-3}$) and dominated by decomposing and fragmented precipitation particles, surface snow may have been subject to metamorphism or melt, creating rounded grains and melt forms which can increase the layer density (100 – 300 kg m$^{-3}$). Depth hoar and wind slab layer properties followed the classifications of Fierz et al. (2009). Spatial variability of layer thicknesses were assessed using uni-directional semi-variograms to estimate correlation length scales of layer thicknesses; horizontal distances over which the thickness of a layer along a trench becomes statistically

uncorrelated. The range to sill of a semi-variogram, using a stable bounded fitting model, was used to quantify this distance for each layer (SS, WS, DH) in all trenches.

In addition to trenches, measurements of the same snow properties were made in 85 snow pits at 54 locations throughout TVC, where each in situ snow measurement was attributed to one of three layers. Whilst only trench data were used in the semi-variogram analysis, data from the trenches and pits in 2013 and the pits in 2018 were combined to determine layer thickness

as a function of snow depth and the microstructural properties of the snow within those layers.

### 2.1.2 Land Surface Slope Classification

To classify the surface topography of TVC by slope position and landform category (Figure 1) a Topographic Position Index (TPI) was calculated following the methods of Weiss (2001). A 1 m resolution digital elevation model (DEM) was used,

created during snow-free conditions in August 2008 (Hopkinson et al., 2011), with an Optech ALTM 3100 lidar. The absolute vertical accuracy of the 1 m DEM was at best ± 25 cm, and the horizontal positional accuracy was ± 50 cm. The TPI values were combined with slope information to classify the study domain into the following landforms: flat upland plateau (<5 degrees), flat valley bottom (< 5 degrees), slopes (>5 degrees), lakes (lake extent was extracted from 1:50,000 Canadian topographic maps).

### 2.1.3 Airborne Lidar Snow Depths

A spatially continuous surface of snow heights in TVC was measured in April 2013 using an airborne laser scanning Riegl LMS-Q240in lidar (Johnson et al., 2013). The lidar, flown at 500 m above ground level, had a laser shot footprint diameter of



~20 cm which was aggregated to a 1 x 1 m surface height product across a swath width of 500 m. Typical measurement errors associated with this system were up to ±20 cm (Johnson et al., 2013). Airborne lidar snow depth was then estimated based on elevation differences between these snow surface heights and the snow-free summer 2008 DEM data (Hopkinson et al., 2011), see section 2.1.2. The 1 x 1 m airborne lidar snow depth raster, was then re-sampled to 10 x 10 m resolution using a cubic

interpolation.

In a subset of all lidar snow depths, King et al. (2018) demonstrated that lidar and in-situ measured snow depths closely agreed in relatively flat, open environments. Lidar and in situ measured mean snow depths in upland tundra areas of TVC were shown to have a RMSE of 8.5 cm (King et al., 2018), while additional unpublished analysis showed a ~14 cm positive bias of lidar snow depths over the whole TVC domain.

## 2.2 SWE retrieval errors using SMRT

The Snow Microwave Radiative Transfer (SMRT) backscatter and emission model (Picard et al., 2018) was used to illustrate potential retrieval error from a dual-Ku band radar system (c.f. King et al., 2018; Lemmetyinen et al., 2018). Up to three layers were assumed within the snowpack: depth hoar (DH), wind slab (WS) and surface snow (SS), with layer thickness dependent

on total snow depth based on relationships derived from snow trenches. Synthetic scenarios were used to simulate 'truth' backscatter of the scene, with information from observed spatial variability in microstructural properties. Horizontally homogeneous snow was assumed for the retrieval, with snow depth retrieved as a function of the estimated snow microstructure. Davenport et al. (2008) used a similar synthetic scene methodology to quantify soil moisture error from passive microwave observations.

For simulation of the truth backscatter, density was held constant for each layer. SSA observations were classified by five equally sized intervals across the observed range. Truth scenes were constructed from five backscatter simulations, aggregated with weights taken from the histogram frequency:

$$\varphi_k^{tru} = \sum_{n=1}^{5} w_n \, \varphi_{n,k} \tag{1}$$

where $\varphi_k^{tru}$ is the aggregated truth backscatter at frequency k, $\varphi_{n,k}$ is the backscatter simulated for a three-layer snowpack with $n$th SSA value. To isolate the impact of individual layers, SSA in two layers were kept spatially constant whilst allowed to vary in the third layer, as illustrated in Figure 3. The weights, $w_n$, derived from the histogram frequencies $f_n$ for each SSA interval ($1 \leq n \leq 5$):

$$w_n = \frac{f_n}{\Sigma f_n} \tag{2}$$

In retrieval snowpack scenes, SSA was assumed constant within layers, as shown by Figure 3. Simulations with a range of first guess estimates of the SSA were used to determine the backscatter of homogeneous scenes and retrieved depth (subsequently converted to SWE) identified as the minimum of a cost function. For these simulations, a simplified version of



the cost function given in Lemmetyinen et al. (2018) was used (Equation 3). From the optimal dual-frequency approach to SWE retrieval determined by Lemmetyinen et al. (2018), the retrieval algorithm in this study was based on the backscatter difference at two frequencies (13.4 GHz and 17.2 GHz at VV polarization) and a fixed incidence angle of 35 degrees. Retrievals followed an equivalent SMRT model configuration to the forward modelling of Lemmetyinen et al. (2018) and King et al.

(2018), i.e. an exponential snow microstructure model and the Improved Born Approximation electromagnetic theory. The model approaches differ in the solution of the radiative transfer equations: a multi-flux solver (Picard et al., 2004; Picard et al., 2014) was used in the SMRT simulations whereas a six-flux solver was used in King et al. (2018).

A perfect retrieval was assumed possible (i.e. negligible noise), and an isothermal temperature of 265K and constant soil backscatter contribution of -13dB was assumed in both truth and retrieval simulations. The assumed soil properties are for

illustration purposes only, given the lack of bare frozen soil backscatter observations at this frequency. This appears to be a plausible value from the snow to snow-free transition period shown in Scipal et al. (2002) and an alternative soil backscatter of -10dB was used to test the robustness of conclusions from the -13dB simulations; however, a full error budget study should consider a range of values. The simplified cost function was:

$$CF(SWE) = \left[\varphi_{diff}^{sim}(d, f(SSA), x_1, \ldots, x_m) - \varphi_{diff}^{tru}\right]^2 \tag{3}$$

where $\varphi_{diff}$ is the backscatter difference $\varphi_{17.2VV} - \varphi_{13.4VV}$, $d$ is snow depth, the estimated microstructure is a function of the SSA, $f(SSA)$, and $x_m$ are other parameters assumed to be known exactly i.e. the same as used in the 'truth' simulations.

## 3. Results

### 3.1 Snowpack Properties

The winter of 2017-18 was warmer than 2012-13 (Figure 4). Similar air temperatures were observed in both winters from

October through November, but December through March in 2017-18 was on average 9°C warmer. Importantly, during 2017-18 there were three short (<1 day) periods where mid-winter air temperatures increased above -5°C and approached melting point. The warm period on 15 January 2018 coincided with reports of light freezing rain at Inuvik airport, the nearest weather observing station 49 km away. Therefore, while similar meteorological conditions influenced early season winter snowpack accumulation and depth hoar metamorphism, a spatially extensive ice lens (>1 mm thick) was formed in the 2017-18 snowpack

because of the January warm event. Ice lenses provide potential to restrict vertical vapour diffusion, although such impacts may be limited due to the additional presence of a dense wind slab layer of already tightly-packed snow grains. Ice lens formation can also restrict the flux of blowing snow, reducing potential for subsequent drifting. Consequently, through the aggregation of snowpack measurements over two winters with strongly differing meteorological conditions, the resulting data set provides a highly valuable description of tundra snowpack properties in a warming Arctic.

Figure 5 compares statistical distributions of snow depths from lidar in three topographically delimited subdomains (Figure 1: flat upper plateau, slope, flat valley bottom) of the TVC catchment. Median snow depths were very similar (0.60-0.65 m)



across all subdomains. The inter-quartile range of snow depths on slopes was largest, reflecting enhanced drift processes that preferentially redistributed SWE to wind-sheltered areas of sloping terrain. However, similarity in frequencies of snow depths greater than 0.9 m between flat upper plateau and sloped subdomains suggested preferential SWE deposition also occurred over relatively flat (<5°) terrain, in addition to drifts in leeward slopes aligned perpendicular to prevailing north-westerly winds

(King et al., 2018). This is of importance as flat upland plateau, where the majority of trenches and pits were located, was the areally dominant subdomain (66% of the total TVC lidar coverage in contrast to 19% slopes and 8% flat valley bottom). Consequently, field measurements well represented both the range and frequency of the total TVC lidar snow distribution. This type of exposed, flat, largely unforested terrain is representative of pan-Arctic tundra environments, allowing potential for implications to be drawn beyond TVC.

Thickness of tundra snowpack layers was spatially variable and frequently laterally discontinuous. Layers expand and contract horizontally, often creating several different discrete entities of the same layer across a trench face (Figure 6). The number of layer entities in snow trenches ranged from 5 to 14 in the 5 m trenches and up to 36 layer entities in the 50 m trench (Table 1). Mean snow depth of trenches ranged between 26 cm and 53 cm for all but trench 6 (79 cm) located in a valley bottom. Consequently, even when considering a positive lidar measurement bias of ~14 cm, trenches were generally located between

the median and lower quartile of the total TVC lidar snow depth distribution (Figure 5). A coherent layer of surface snow was evident in four trenches (Table 1), consisting of up to 26% of the mean layer thickness. The proportions of wind slab layers (35 to 80%) and depth hoar layers (20 to 46%) exhibited a larger range. Figure 7 shows the mean proportion of depth hoar was consistently just under 30% of total snow depth. The mean proportion of wind slab was consistently greater than 50% and showed an increasing trend with increasing total snow depth, indicating that (other factors being equal) where wind slab

thickness was greater, so was the total depth of the snowpack. A decreasing trend in the mean proportion of surface snow (approximately 25% to 0%) with increasing total depth was most likely a result of greater wind erosion and re-distribution from the surface where the snowpack was deeper and more wind affected. While interquartile ranges around these trendlines express the natural variability in measured proportional thickness, where total snow depth is known, trendlines made it possible to estimate the percentage of wind slab and depth hoar in a snowpack of unknown microstructure, thereby allowing potential

for application of these relationships over larger spatial scales.

Centimetre-scale variability in tundra snowpack layer thickness was quantified from trench measurements in a spatially distributed manner throughout a catchment for the first time. The range to sill (i.e. horizontal correlation length) of semi-variograms (Figure 8) was used to quantify spatial variability, which varied for all layer thicknesses between 16 cm and 158 cm (Table 2). While the semi-variance of layer thickness in trenches (Figure 8) changed as a function of absolute layer

thickness (Table 1), the mean range to sill of layer thickness increased from depth hoar (45 cm), to wind slab (59 cm), to surface snow (81 cm). The mean range to sill of total snow depths (61 cm) was only slightly greater than that of wind slab, suggesting horizontal variability in wind slab thickness has a strong control over total snowpack thickness. Subnivean roughness, the boundary between snow and the underlying substrate, is likely to have a strong influence on depth hoar thickness. To estimate the importance of this influence, roughness was quantified as twice the value of the root mean square



of residuals between the snow-substrate boundary and a linear best fit line to that boundary (Table 1). This provides an estimate of the peak to trough amplitude between adjacent subnivean topographic features, often controlled in tundra environments by tussock grasses (e.g. Figure 6). The roughness metric was calculated across 2 m moving windows. The starting position of each window moved in 1 cm horizontal increments along each trench and then the roughness of all moving windows were

averaged per trench. The 2 m distance was chosen to exceed the maximum measured horizontal correlation length of layer thicknesses, while also being broadly representative from visual field inspection of the spacing between tussocks. Across all trenches, roughness of the subnivean boundary ranged from 9-32% of total trench depth, with a mean of 18% for all trenches. This is consistent with the premise that depth hoar layer thickness (~30% of total snowpack depth) is strongly influenced, but not exclusively controlled, by subnivean roughness.

Figure 9 demonstrates relationships between mean percentage thickness of different layers, density and SSA from a combination of snow microstructural profiles from trenches and pits. Layers were primarily delimited in the field by vertical profiling (visual inspection and hardness). Figure 9a and b show median densities of 104 kg m$^{-3}$ (surface snow), 253 kg m$^{-3}$ (depth hoar), and 316 kg m$^{-3}$ (wind slab); differences in median densities between layers were greater than the 5-9% sampling error associated with gravimetric cutters (Proksch et al., 2016). The inter-quartile ranges of each layer density did not overlap

indicating clear differences between layer densities, even though there was overlap between full measurement ranges. The upper quartile of surface snow densities overlapped densities of both wind slab and depth hoar, as although it is structurally distinct from the lower wind slab layer, surface snow may have been subject to decomposition, melt or some wind-packing effects. Additionally, overlap between the lower quartile of wind slab densities and the upper quartile of depth hoar densities, resulted from densities in lower sections of wind slab which exhibited the hardness of wind slab yet also microstructural similarity to depth hoar. This is often reported in Arctic tundra snowpacks that undergo strong temperature gradient

similarity to depth hoar. This is often reported in Arctic tundra snowpacks that undergo strong temperature gradient metamorphism, and has previously been classified as a unique layer type such as indurated hoar (Sturm et al., 1997; Fierz et al., 2009; Derksen et al., 2014; Domine et al., 2016). Differences between SSA of different layers were more distinct than for densities (Figure 9c and d); inter-quartile ranges did not overlap as median SSA increased from depth hoar (11 m$^2$ kg$^{-1}$), to wind slab (24 m$^2$ kg$^{-1}$), to surface snow (45 m$^2$ kg$^{-1}$). Differences in median SSA between layers were much greater than the

10% measurement uncertainties of SSA (for snow <60 m$^2$ kg$^{-1}$) from IR hemispherical reflectance techniques at 1310 nm (Gallet et al., 2009). Consequently, as the density and SSA values of each layer are nicely separate from each other, it is reasonable to expect that with respect to radar backscatter, it is the relative proportions of these snow components, and their attributes, which will drive the radar results. In the next section we explore this using a three-layer radiative transfer model.

## 3.2 SWE retrieval accuracy from radar backscatter

Notable differences between distributions of SSA and density in different snowpack layers allowed parametrisation of snowpack microstructure in the SMRT model (Table 3). Coupled with fitted relationships between total snow depth and layer thickness as a proportion of total depth (Table 3 and dotted trendlines in Figure 7) SWE retrieval errors from SMRT simulations were calculated as a function of measured variability in SSA in different layers. SSA measurements of snowpack layers show



positively skewed distributions (Figure 10 a, c and e); surface snow, wind slab and depth hoar distributions comprised 64, 77 and 85 averaged layer measurements respectively. SWE retrieval errors were calculated as truth SWE minus retrieved SWE. SWE retrieval error due to heterogeneity in SSA for SS, WS and DH layers is presented in Figures 9b, 9d and 9f for snowpacks of depths between 0.2 and 1m, and for estimates of layer SSA within 20% of the known median value.

Spatial variability in surface snow SSA has negligible effect on the retrieval error (Figure 10b). Instead, the retrieval is most sensitive to spatial variability in depth hoar (Figure 10f) despite only a small range in measured SSA values. A 20% underestimation in SSA leads to an underestimation in retrieved SWE by as much as 100 mm SWE in 0.8 m deep snowpacks; an underestimation of between one- and two-thirds of total SWE assuming typical tundra snow densities. Overestimation of SSA leads to nearly as large an error in the other direction. Errors in assigning SSA values to wind slab also affect the SWE

retrieval, but to a much lesser extent. While a perfect retrieval is possible at all depths (shown by the white contour line in Figure 10f) an estimated SSA lower than the median is required. For depth hoar, approximately 7% less than the median is required. For wind slab, closer to 10% under the median is required.

A radar SWE retrieval algorithm will have constraints on the allowable accuracy. Here, we have set a limit of ±30 mm SWE, indicated as black contour lines in Figure 10 as the acceptable limit on SWE. This error constrains the setting of SSA values

(black contour lines in Figure 10f), which become increasingly stringent for deeper snow. The constraints in SSA are considerably more liberal for wind slab (Figure 10d) than for depth hoar as the retrievals are less sensitive to the spatial variability in SSA. Figure 10f (DH) shows that as snow depth exceeds 0.6 m, the need for accurate values of SSA for depth hoar become critical. If the median SSA is used for depth hoar, the maximum SWE error is 40% of the truth SWE. For depth hoar SSA 20% above the median, the SWE retrieval error can exceed the actual SWE, particularly for shallow snowpacks. A

higher soil backscatter contribution of -10dB does not change the optimal SSA estimates, but does result in more stringent retrieval requirements. For example, at 60 cm snow depth the SSA should be between 89 and 97% of the median value to remain within error budget for soil backscatter of -13dB. For a soil backscatter of -10dB this range is reduced to between 90 and 96%. This highlights the need for field observations of backscatter from bare frozen soil to better constrain this value.

Estimates of the single SSA of layers in retrieval scenes will always be lower than the median because of non-linearity between

scattering and SSA. Snow layers with smaller SSA will have a disproportionately larger effect in the truth scene as these microstructure elements will scatter more (optical grain diameter is inversely proportional to SSA). SMRT simulations can compensate for the heterogeneity by assuming slightly smaller SSA in the homogeneous scene. The amount of compensation required could differ as a function of ground contributions, but the representative SSA will always be smaller.

## 4. Discussion and Conclusions

Comparison between the mean snow depth of all trenches (40 cm) and distributions of lidar-derived snow depths demonstrated that trenches were highly representative of snow depths across the whole TVC catchment, incorporating a range of different topographic landscape units. Snow depths in trenches were also consistent with snowpacks found over much wider spatial



scales; mean snow depths of 39 cm and 23 cm for sub-Arctic and Arctic snowpacks respectively were reported by Derksen et al. (2014). In addition, trenches at TVC were highly representative of the previously documented complex and often discontinuous layering of tundra and sub-Arctic snowpacks (Benson and Sturm, 1993; Sturm and Benson, 2004; Domine et al., 2012; Rutter et al., 2014; Rutter et al., 2016). Consequently, through the coincident combination of spatially distributed
measurements of snow microstructure (94 profiles at 36 sites across two winters) and snow layer thickness (9000 vertical profiles across 9 trenches during one winter) we present a unique and robust data set that is likely to be representative of tundra snowpacks in general. The application of these data in a snow microwave radiative transfer model (Figure 10) therefore is of high potential relevance to remote sensing of seasonal snow in the Northern Hemisphere.

Horizontal correlation lengths of layer thicknesses (i.e. distances over which variability in the thickness of individual layers
decouples and becomes independent) have not previously been reported. High resolution layer boundary identification using stitched and georeferenced NIR imagery (Tape et al., 2010; Watts, 2015) now allows spatial variability of layer thickness to be quantified using semi-variograms in analytical approaches similar to assessment of snow depth distributions (e.g. Deems, 2008; Trujillo 2015). Horizontal correlation lengths (i.e. range to sill of semi-variograms) and variance of major layer thicknesses at TVC suggests depth hoar layers vary the most over the shortest distances, because of the subnivean vegetation
and topography. In tundra environments, especially on topographically exposed plateaus, undulating subnivean topography traps early winter snowfall in troughs between tussocks. Discontinuities in snowpack layering occur between adjacent ridges and troughs (often caused by tussock grasses) and trapped snow is then subject to strong temperature gradient metamorphism (Colbeck, 1983; Colbeck, 1987), creating large depth hoar crystals. The decreasing spatial variability (increasing distance in horizontal correlation length) of wind slab and surface snow layer thickness reflects the decreasing influence of topography
once snow has filled local hollows and the increasing predominance of spatially smoother wind-driven processes such as snow redistribution and compaction. For comparison, the horizontal correlation length of total snowpack thickness, a consequence of variability in all three layers, approximates that of wind slab. Quantification of layer thickness variability using trench measurements suggests a relative hierarchy of variability in commonly occurring tundra snowpacks layer types, which all vary significantly at the sub-metre scale.

Correlation lengths highlight minimum measurement distances over which sampling of snowpack properties should take place. While the intensity of such sampling may have practical limitations, targeted sub-metre sampling as part of wider ground-based, catchment-scale snow measurement campaigns is highly relevant for evaluation of current and future satellite sensors that operate over resolutions on the scale of metres to tens of metres (e.g. Cline et al., 2009; Yueh et al., 2009; King et al., 2018). Rapid acquisition of vertical profiles of snowpack properties using snow micropenetrometers (Schneebeli et al., 1999;
Proksch et al., 2015) may increasingly provide the enhanced field measurement capacity required to achieve this required spatial sampling resolution.

Correlation lengths can be applied in distributed modelling of snowpack properties using kriging techniques to enable spatial interpolation. Application of the length scales of major snowpack layers, as well as variability of properties within each layer, has potential for use in catchment scale models along with knowledge of other landscape elements such as snow drifts. While



accurate snowpack modelling at the tens of metres scale in tundra environments is challenging (Essery et al., 1999; Clark et al., 2011), snowpack layer correlation lengths could be used to parametrise models so variability of snowpack properties are reliably accounted for when modelling at coarser resolutions. This will become an important parameterisation for future linkages between physical snowpack models and one-dimensional radiative transfer models (Sandells et al., 2017), to better

model surface and volume scattering in snow (Zhu et al., 2018), and in observing system simulation experiments (e.g. Garnaud et al., 2019) to assess the performance of future Ku-band radar satellite sensor configurations.

Relationships between layer thicknesses and total snow depth are presented for purposes of simplifying snowpack layer complexity for applications over large scales. Linear trendlines provide a guide to changes in layer proportions for snowpacks up to 1 m deep. Interestingly, the proportion of depth hoar is consistently ~30% irrespective of total depth, a trend which

continued in the few snow pits in 2018 that were deeper than 1 m. This is somewhat surprising, due to the previously described strong controls of ground surface roughness over depth hoar creation and the potential for large wind slabs to then dominate as the total snow depth increases above the top of ridges. However, the proportion of wind slab layer instead increases at the expense of the decreasing surface snow layer. Consequently, this proportional approach to layer partitioning of snowpacks provides an alternative to applying a maximum depth hoar thickness of 25 cm (King et al., 2018), and therefore may be more

appropriate for wider applications across pan-Arctic tundra over multiple winter seasons.

Although rates of compaction, and hence density, of a dry snow layer under its own mass will have a proportional relationship to its thickness (Colbeck, 1972; Anderson, 1976) the weak relationship for fresh surface tundra snow is expected due to other influential aeolian and metamorphic processes. In addition, there was unlikely to be a direct relationship between density and proportional layer thickness for either wind slab or depth hoar, due to wind and thermal gradients dominating processes which

control the density of these respective layers. Such processes are currently very challenging to simulate accurately in Arctic snowpacks (Barrere et al., 2017; Domine et al., 2019). So, as distinct differences are evident between distributions of layer properties, using a single median value to represent the density for each layer is appropriate. Differences are even more profound between distributions of SSA in different layers which exhibit low inter-quartile ranges, especially for depth hoar layers which dominate Ku-band microwave scattering (King et al., 2015). This is particularly important for depth hoar as non-

linear microwave scattering with respect to SSA (SSA is inversely proportional to exponential correlation length or optical grain diameter) means that the smaller SSA will have a disproportionate scattering effect compared with larger SSA. This is demonstrated by synthetic experiments that evaluate the impact of spatial uncertainty in snow microstructural parameters on SWE retrievals, where a single SSA describing a homogeneous scene must be smaller than the median of a heterogeneous scene to compensate for the non-linear scattering response. Small SSA corresponds to high scattering so it follows that an

underestimation in the SSA estimate (too much scattering) results in less scattering material needed (an underestimation in the retrieved SWE). Whilst the effects of spatial variability in SSA can be countered, the value of representative SSA is critical. Although the SWE retrieval accuracy requirements for depth hoar are more stringent than for wind slab, it is known more precisely than wind slab SSA for the TVC data (Figure 9). The insensitivity of the retrieval to the spatial variability of surface





snow SSA suggests that a median value of SSA will suffice for SWE retrievals. Future work will consider whether a two layer retrieval system could be used to represent the scattering of a three-layer snowpack.

By applying a comprehensive data set of snow microstructural properties collected over two winters at TVC in synthetic radiative transfer experiments, the impact of first guess information of snow microstructural characteristics on the viability of

SWE retrieval strategies relevant for satellite mission design have been robustly assessed. Spatial variability in microstructure of depth hoar layers dominates SWE retrieval errors. A depth hoar SSA estimate of around 7% under the median value is needed to accurately retrieve SWE for this snow. In shallow snowpacks <0.6m, depth hoar SSA estimates of ±5-10% around this value allow retrievals within a tolerance of ±30 mm SWE. Where snowpacks are deeper than around 30cm, accurate values of representative SSA for depth hoar become critical as the retrieval error will be exceeded if the median depth hoar SSA is

applied. Importantly, these experimental results allow potential for uncertainty in SSA in Arctic tundra snow to be used to produce future SWE retrieval quality flags in remotely sensed products, and also provide benchmark accuracies for physical snowpack models to deliver SSA estimates. As modelling microwave scattering in large scale applications has currently been limited to single layer snowpacks (Takala et al., 2011), the potential for using multilayer information is exciting and progressive. Especially when considering the strongly different scattering properties of wind slab and depth hoar (King et al.,

2018), the combined influence of which dominates control over microwave scattering in Arctic tundra snowpacks.

**Data Availability**. Field data are publically available using the following links:

Snow pits and trenches (https://doi.org/10.6084/m9.figshare.8397737.v2)

Lidar snow depths (https://doi.org/10.6084/m9.figshare.8397467.v1)

Air temperatures (https://doi.org/10.6084/m9.figshare.8397419.v1)

Catchment topography and locations of snow pits / trenches (https://doi.org/10.6084/m9.figshare.8397023.v1)

Code and data to generate the SMRT outputs are available at https://github.com/mjsandells/Rutter_TVC_2019.

**Author contributions**. NR led trench data collection, analysis of field data and manuscript preparation, MJS led SMRT analysis and contributed to data analysis and manuscript preparation, CD coordinated TVC ground and airborne field

measurement campaigns, PT led topographic analysis of TVC terrain, CL led lidar snow depth data collection. LW contributed to spatial analysis of snow depth and snowpack layering. TW led trench data collection and contributed to data analysis. NR, CD, JK, PT, TW, RE, A Roy, A Royer, PM and MS all made significant contributions to collection of field data and assisted in paper preparation.

**Competing interests**. The authors declare that they have no conflict of interest.

**Acknowledgements**. Data collection and analysis were supported by the European Space Agency (SCADAS: 4000118400/16/NL/FF/gp), University of Alaska Fairbanks, Canadian Space Agency (13MOA07103), Environment and Climate Change Canada, Northumbria RDF Studentship (to TW), and NERC Arctic Office UK & Canada Arctic Partnership Bursaries Programme (to NR and RE).



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



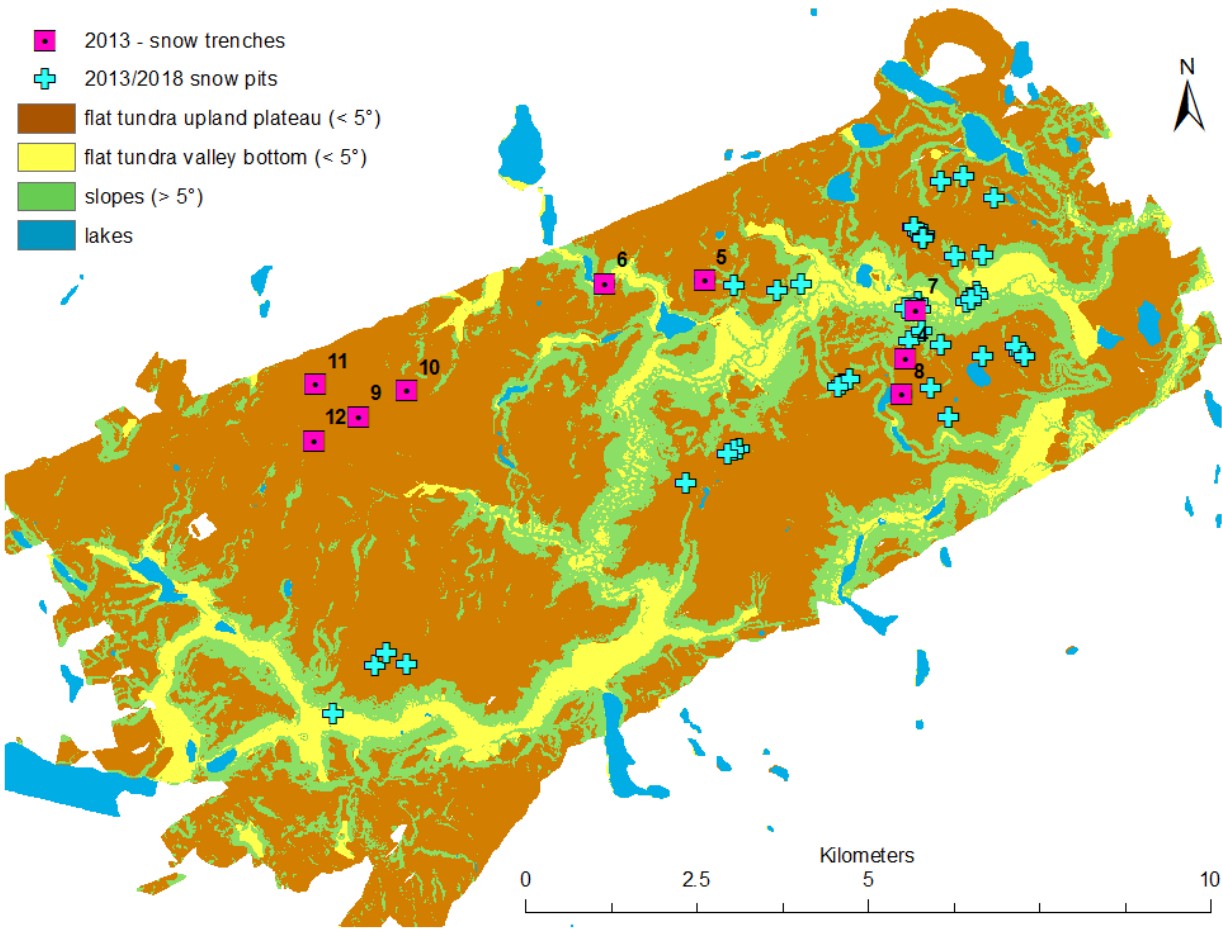

**Figure 1: Locations of snow trenches excavated in April 2013 are indicated by the pink squares. Locations of snow pits excavated in April 2013 and March 2018 are indicated by the blue crosses. Brown, yellow, green and blue colours on the map denote the different terrain types identified by the topographic position index. In 2018 five additional pits were located on Husky lakes (not on map) approximately 5km to the east of the presented domain.**



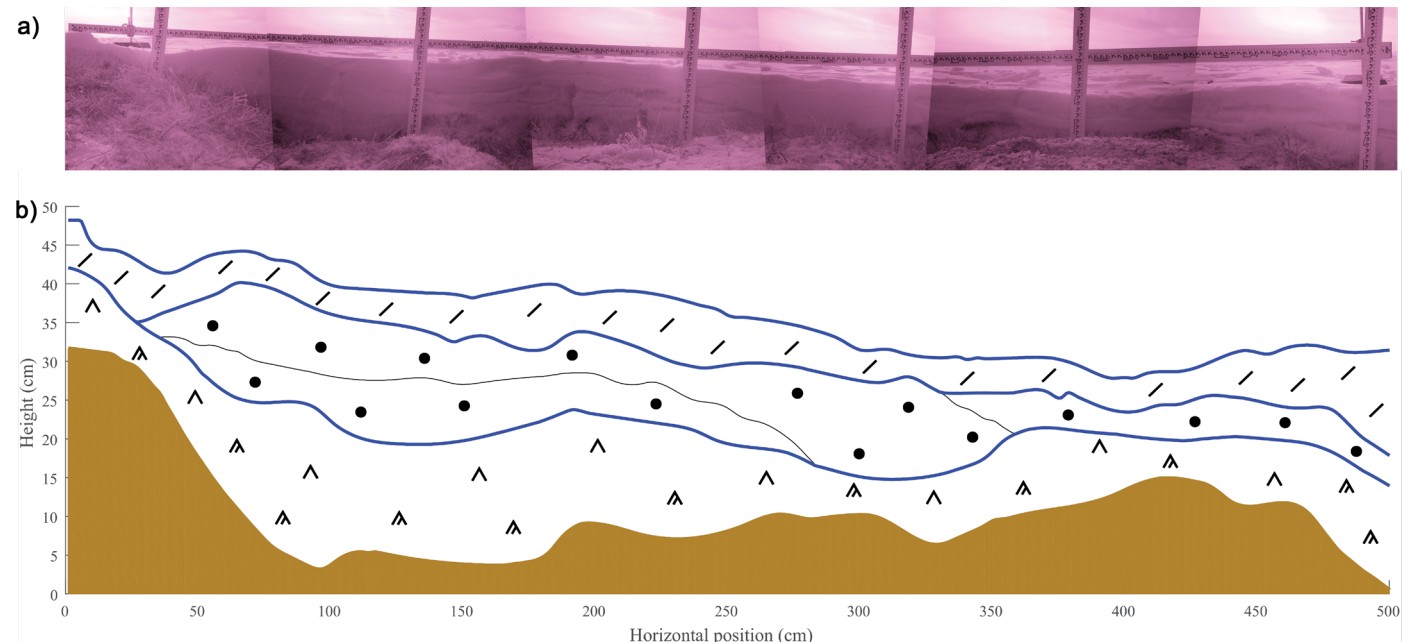

**Figure 2: a) Stitched NIR images of the trench face (Trench 10), b) Layer boundaries derived from NIR imagery, blue lines highlight boundaries between snow and air , surface snow and wind slab, as well as wind slab and depth hoar. The brown area is subnivean soil or vegetation. Symbols describe snow type following the classification of Fierz et al. (2009).**





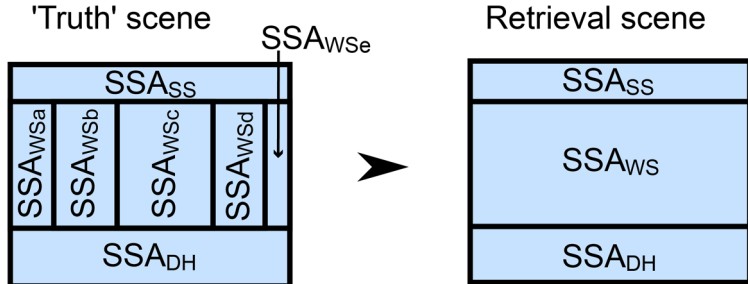

**Figure 3: Illustration of three-layer (SS: Surface Snow; WS: Wind Slab; DH: Depth Hoar) truth scene with spatial variability in Specific Surface Area of WS (SSA$_{WSa}$ to SSA$_{WSe}$) and the retrieval scene with horizontally uniform SSA. Spatial distribution in truth SSA given by observed spatial distribution at TVC over two winters (2013 and 2018).**

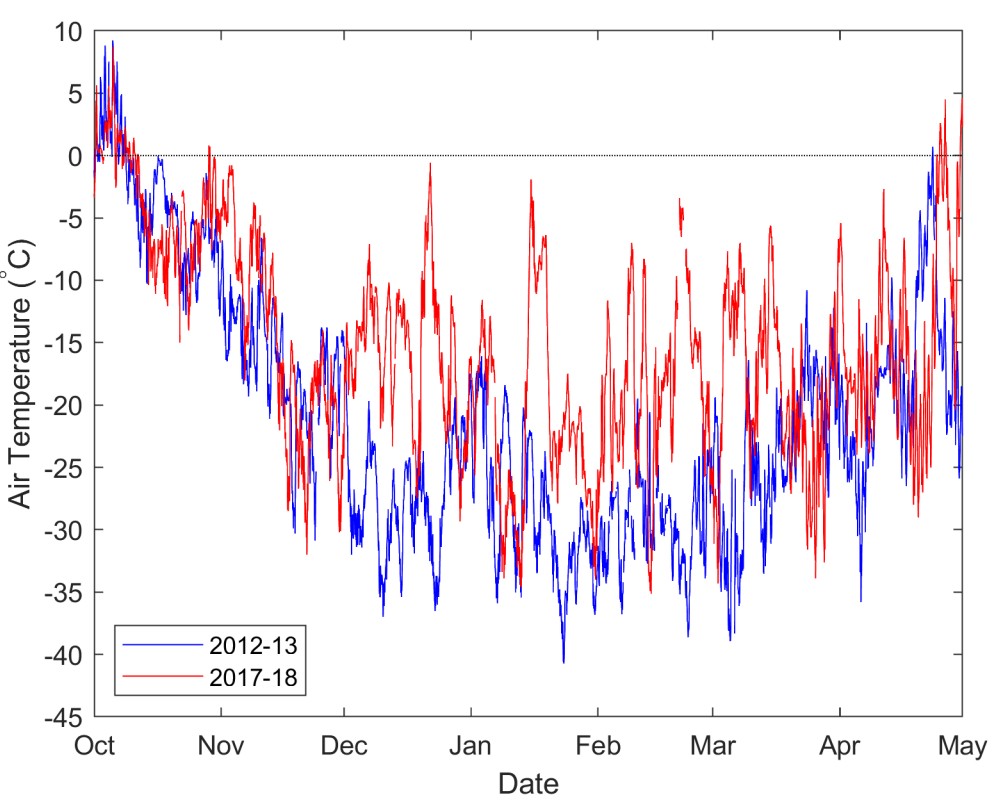

**Figure 4: Air temperature at Trail Valley Creek during winter 2012-13 and 2017-18.**



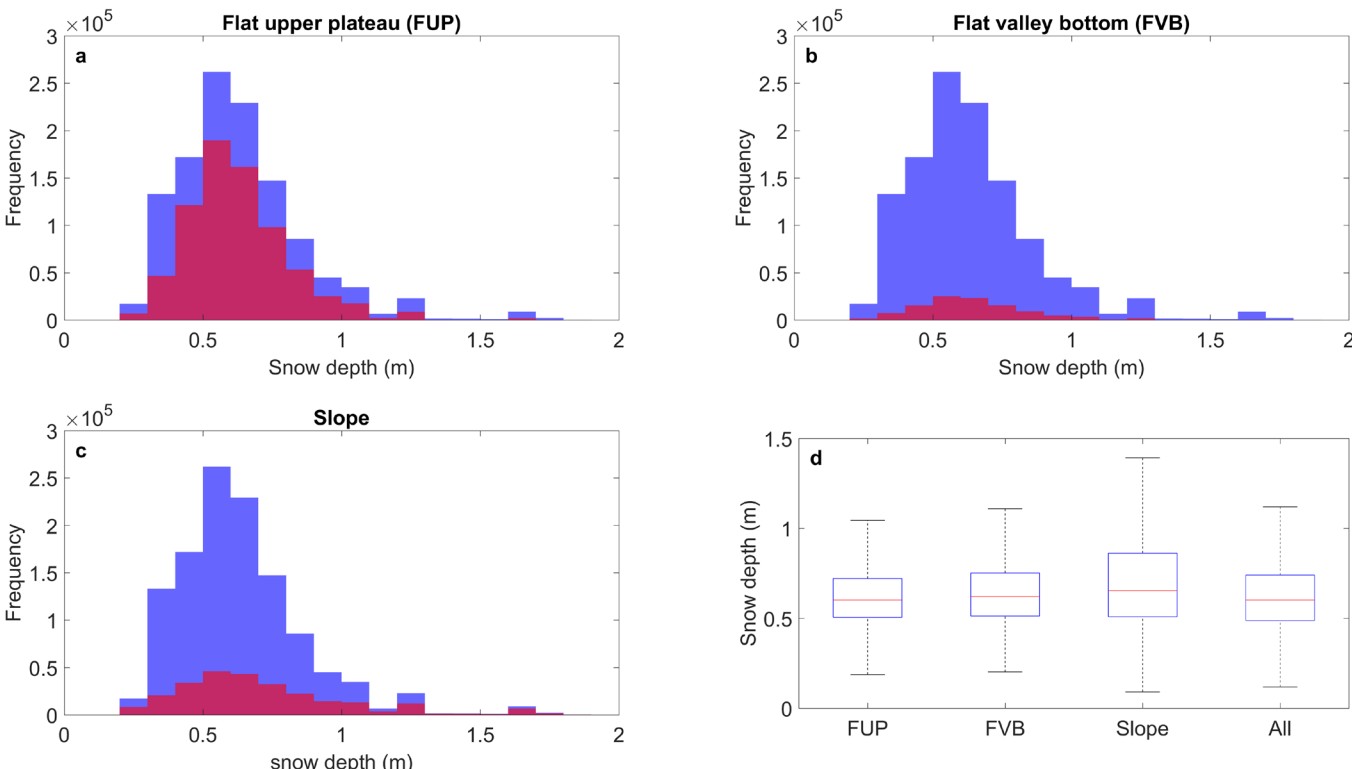

**Figure 5: Snow depths (limited to 2m) in TVC from airborne lidar: a) to c) histograms of individual land surface types (red) overlaid on the histogram of all snow depths (blue); d) distributions of snow depth by land surface type: blue box (inter-quartile range), red line (median), whiskers (dashed lines) extend from the end of each box to 1.5 times the interquartile range, outliers beyond this range are omitted.**

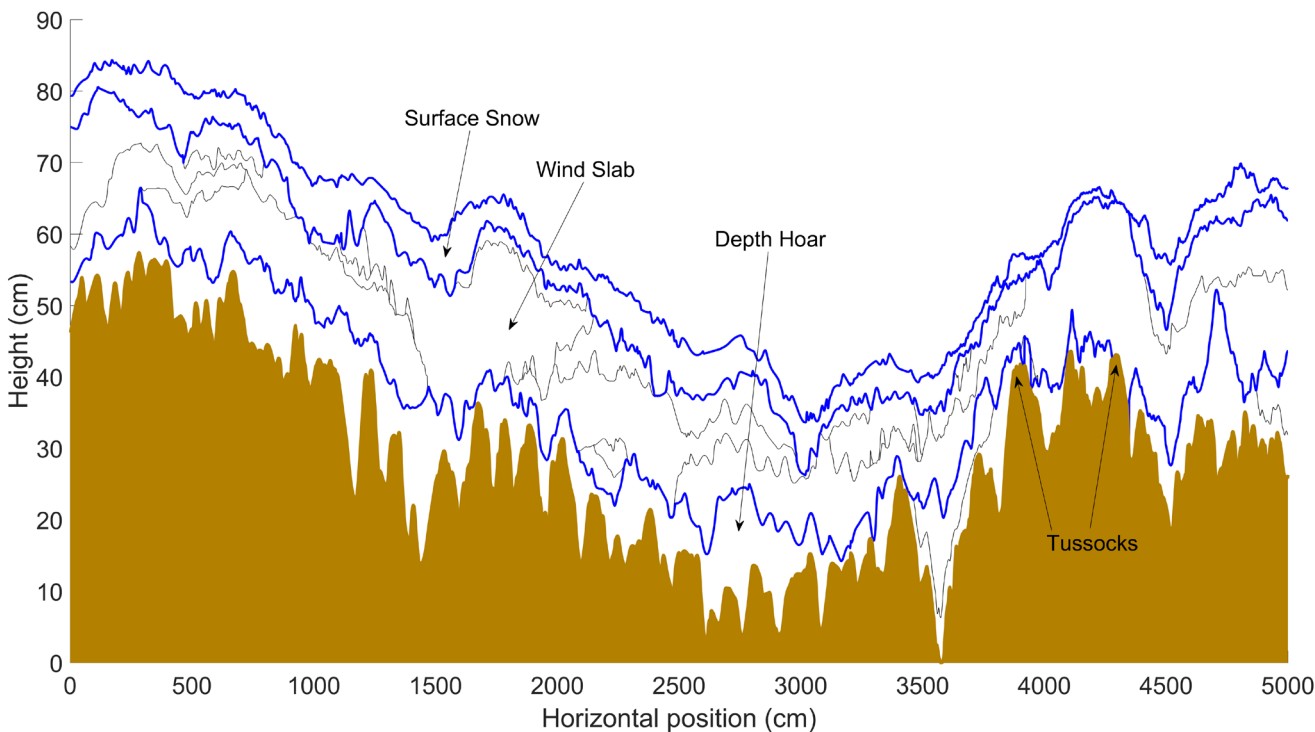

**Figure 6: Cross-section with vertical exaggeration of layer boundaries of individual stratigraphic layers in trench 4 (50 m). Blue lines highlight boundaries between snow and air, surface snow and wind slab, as well as wind slab and depth hoar. The brown area is subnivean soil or vegetation (two tussoscks are labelled). Black lines show boundaries of individual layers aggregated within the wind slab and depth hoar layers.**







**Figure 7: Relative change in thickness of snowpack layers with total depth: median (solid) and interquartile range (shaded) of Surface Snow (red), Wind Slab (blue) and Depth Hoar (black) layers. Dotted line describes the dependence of layer thickness on depth based on linear trendline fits. Layer thickness data from 2013 trenches only.**



**Figure 8: Semi-variograms in individual trenches for a) total snow depth, and layer thickness for b) Surface Snow, c) Wind Slab, and d) Depth Hoar.**



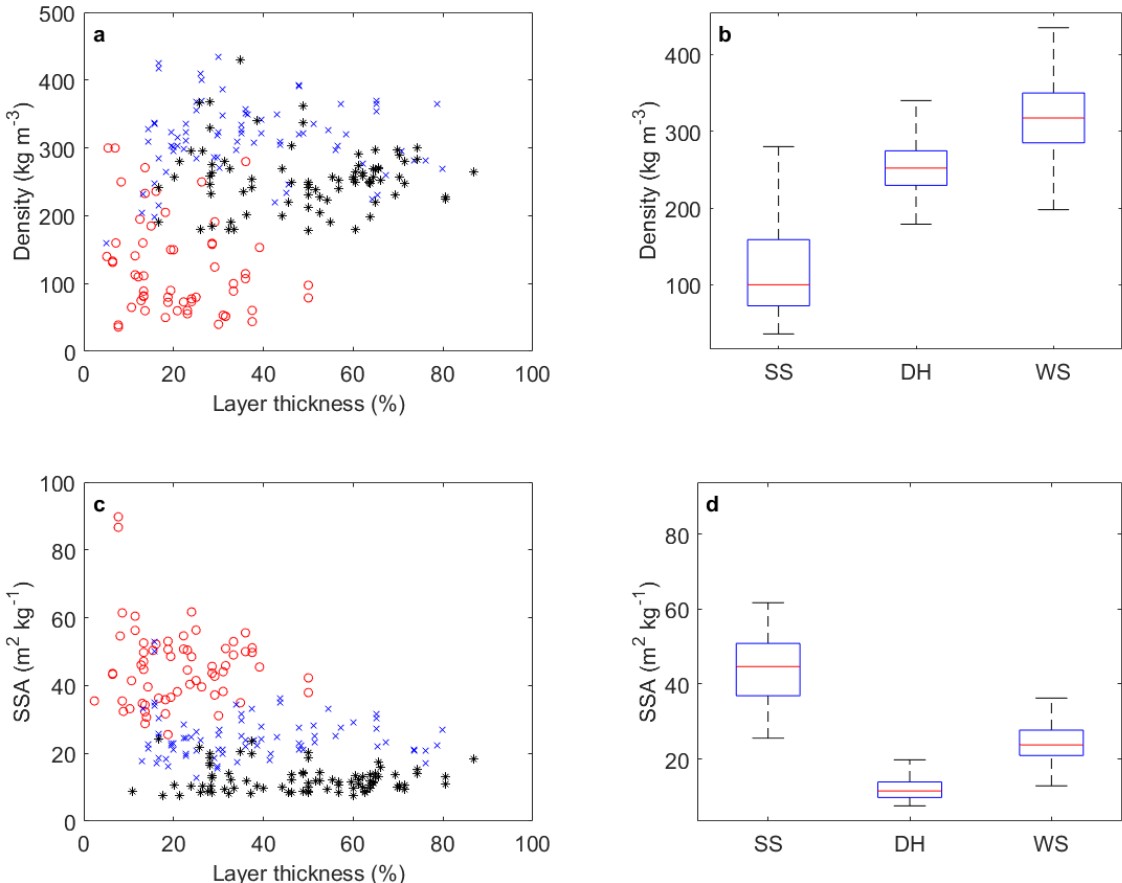

**Figure 9: Change in snow layer a) density and c) SSA with relative thickness of snowpack layers: Surface Snow (red circle), Wind Slab (blue cross) and Depth Hoar (black star). Distributions of snow layer b) density and d) SSA: blue box (inter-quartile range), red line (median), whiskers (dashed lines) extend from the end of each box to 1.5 times the interquartile range, outliers beyond this range are omitted.**

**Figure 10: Left: Histogram of SSA within each layer. Right: Synthetic error budget study for a range of snow depths assuming homogeneity in the retrieval. Contour lines show acceptable SSA error as a function of snow depth to remain within a ±30 mm SWE retrieval accuracy limit. White contour line shows perfect retrieval.**



| Trench | Length (m) | Number of layer entities | Mean snow depth (cm) | Subnivean roughness (cm) | Mean layer thickness (and % of total thickness) | | | | | | Mean layer density (kg m⁻³) | | | Mean layer SSA (m² kg⁻¹) | | | Local topographic gradient |
|---|---|---|---|---|---|---|---|---|---|---|---|---|---|---|---|---|---|
| | | | | | SS (cm) | SS (%) | WS (cm) | WS (%) | DH (cm) | DH (%) | SS | WS | DH | SS | WS | DH | |
| trench4 | 50 | 36 | 31 | 6 | 5 | 16 | 17 | 55 | 9 | 29 | 240 | 300 | 230 | 52 | 25 | 10 | Flat Upland Tundra |
| trench5 | 5 | 6 | 32 | 3 | - | - | 17 | 54 | 15 | 46 | - | 310 | 220 | - | 39 | 8 | Flat Upland Tundra |
| trench6 | 5 | 14 | 79 | 7 | - | - | 58 | 73 | 21 | 27 | - | 280 | 300 | - | 21 | 10 | Flat Valley Bottom |
| trench7 | 5 | 7 | 29 | 9 | 8 | 26 | 14 | 48 | 7 | 26 | 250 | 320 | 180 | 40 | 28 | 9 | Flat Valley Bottom |
| trench8 | 5 | 5 | 30 | 6 | - | - | 23 | 78 | 6 | 22 | - | 370 | 280 | - | 22 | 8 | Flat Upland Tundra |
| trench9 | 5 | 10 | 34 | 10 | 1 | 2 | 22 | 64 | 11 | 33 | - | 230 | 180 | 35 | 22 | 14 | Flat Upland Tundra |
| trench10 | 5 | 5 | 26 | 4 | 6 | 22 | 9 | 35 | 11 | 44 | - | - | 200 | 40 | - | 10 | Flat Upland Tundra |
| trench11 | 5 | 8 | 43 | 4 | - | - | 29 | 67 | 14 | 33 | - | 260 | 190 | - | 23 | 12 | Flat Upland Tundraa |
| trench12 | 5 | 7 | 53 | 7 | - | - | 42 | 80 | 11 | 20 | - | 270 | 260 | - | 27 | 11 | Flat Upland Tundra |

Table 1: Descriptive statistics of snow trenches excavated in April 2013. See text for explanation of subnivean roughness metric. For thickness statistics, individual layers were aggregated into three common types: SS = Surface Snow, WS = Wind Slab, DH = Depth Hoar (n.b. trenches 1-3 were discarded from analysis due to measurement error).





|  | DH | WS | SS | Total depth |
|---|---|---|---|---|
| **trench4** | 37 | 79 | 77 | 42 |
| **trench5** | 44 | 29 | - | 64 |
| **trench6** | 62 | 137 | - | 138 |
| **trench7** | 16 | 53 | 44 | 31 |
| **trench8** | 18 | 19 | - | 20 |
| **trench9** | 26 | 100 | 158 | 41 |
| **trench10** | 117 | 46 | 44 | 122 |
| **trench11** | 56 | 35 | - | 70 |
| **trench12** | 29 | 30 | - | 24 |
| **mean** | 45 | 59 | 81 | 61 |

**Table 2: Mean range to sill (in cm) for thicknesses of each layer (Surface Snow, Wind Slab, Depth Hoar) and total snow depth using the Stable variogram model.**



| | Snowpack Truth Scene | | |
| --- | --- | --- | --- |
| | Surface Snow | Wind Slab | Depth Hoar |
| Thickness $\Delta z$ (% depth) | max(0, -44.7269 * depth(cm) + 30.1551) | $1 - \Delta z^{SS} - \Delta z^{DH}$ | 29.6 |
| Density(kg m$^{-2}$) | 103.7 | 315.5 | 253.1 |
| Fixed SSA (m$^2$ kg$^{-1}$) | 44.7 | 23.8 | 11.5 |
| Assumed Temperature (K) | 265 | 265 | 265 |

**Table 3: SMRT parameters derived from TVC data. Thickness relationship derived from NIR-derived stratigraphy observed in 2013. Density and fixed SSA taken from median of all observations for that layer type in 2013 and 2018. Fixed SSA were used in place of the SSA distribution when assessing the impact of spatial variability in other layers.**