# Peer review of "Effect of snow microstructure variability on Ku-band radar snow water equivalent retrievals"

_The Cryosphere, 2019_

## Referee Comment (RC1) · Anonymous Referee #1 · 26 Aug 2019

General comments

The authors present a well-written and well-reasoned study into the effects of snow microstructure on Ku-band SWE retrieval in sub-Arctic and Arctic landscapes.

Please define the acronyms SWE and SSA at their first use in the Introduction.

Detailed comments

Figure 5: What are the "histograms of individual land surface types (red)"? Please clarify. These seem not to be discussed in the text either.

P7, L11: What is a layer "entity"? Please define/describe. Their number is indicated in Table 1, suggesting that they are a significant element; however, they do not seem to

be discussed in the text.

P9, L3: Reference to Figures 9x should be 10x.

P9, L11: Should this be "...Figure 10 d and f)..."? Also, should Figure 10b have a white line?

P11, L23: Although the use of single median values for density is reasonable, and density is not the focus of the paper, please comment on the relative effects that density may have on SWE retrieval error. That is, what would Figures 10b,d,f look like if density were to be perturbed within their interquartile ranges?

In the Discussion, please also comment on the likely influence that ice lenses (such as the one mentioned on Page 6, Line 24 for 2017/18) may have on SWE retrieval error.

---

## Referee Comment (RC2) · Anonymous Referee #2 · 30 Aug 2019

Review of "Effect of snow microstructure variability on Ku-band radar snow water equivalent retrievals" by N Rutter et al.

The authors present a really nice study of the effects of horizontal spatial variability of snow microstructure on retrieval errors. The study is in two parts: first a detailed description of snowpack properties measured in Trail Valley Creek, and second a synthetic retrieval experiment, where synthetic radar observations are generated using a radiative transfer model forced by realistic snow stratigraphy, and then those synthetic observations are processed with a retrieval algorithm. They show that SSA needs to be known quite precisely a priori in order to hit the target accuracy requirement.

The first part is wonderful: it shows in depth the spatial variability of observed properties, and occupies nearly all of the figures and tables. The second part I found hard to

understand, so am asking for clarifications. To the extent I understand it, I think it is a critical contribution to this field!

In such studies, the details of the synthetic experiment design can make a lot of difference. After multiple re-reads, I had some trouble piecing together what exactly was done. All of my minor comments below are requests for clarification.

My understanding is that this study performs a set of idealized depth retrieval experiments in order to isolate the impact of a singe phenomenon (spatial variability of SSA in a single layer) on retrieval accuracy. It assumes (in my understanding) 1) perfect radar observations; 2) perfect knowledge of snow density; 3) perfect knowledge of background (i.e. soil and other substrate properties) 4) perfect ability to transform SSA into exponential autocorrelation length, and 5) perfect knowledge of SSA in two of the three layers in the snowpack. My understanding is that only depth is estimated by the retrieval, then transformed to SWE; SSA is assumed to be given. The study then systematically varies a spatially homogenous SSA value provided as a constant to the retrieval, and estimates depth, transforms to SWE, in order to compute the error metrics shown in Figure 10.

Anyway please clarify these minor points! I look forward to reading a revised version.

Minor Comments 1. Page 5, line 11. Please clarify somewhere that density is assumed to be known, i.e. it is not being estimated by the retrieval algorithm, and you are giving the radar simulations for the "retrieval scene" the true density. 2. Page 5, line 11. Please clarify somewhere that SSA is treated as a specified input in the retrieval, if that is the case. I'm assuming that it is treated as "fixed" in the retrieval, in other words, you systematically specify a range of values, but the retrieval algorithm is not actually trying to estimate it. I'm also assuming that for each "layer" experiment, SSA in one layer is treated as spatially variable in the truth (using eqn 1), and is varied systematically in the retrieval scene (as shown in Figure 10), but that the other two layers are not only treated as spatially homogenous in the truth, but are also the "retrieval" simulations are

given the true value of SSA. Please clarify this! I've read through a number of times but cannot find that information. 3. Page 5, line 13-14. "Up to three layers were assumed within the snowpack". Can you reword this? I found it really confusing. 4. Page 5, line 16. "Horizontally homogenous snow was assumed for the retrieval". Please just clarify more explicitly here that you consider horizontal spatial variability in the truth, but horizontal spatial homogeneity in the retrieval. 5. Page 5, line 20-21. Please say why you chose 5 intervals? Readers may assume this maps to "landforms" described in page 4, lines 27-29. I assume 5 is more or less arbitrary, or minimum needed to capture spatial observed distributions, which is fine, but please clarify. Also, this is a great chance to explicitly say that the set of 5 simulations represent spatial variability in that the frequency and weights represent the proportion of a scene that might take each SSA value. 6. Page 5, line 21. "across the observed range". This is referring to the in situ datasets of SSA, correct? But readers could easily get confused as this is how you are computing the synthetic radar observation. Please clarify the language? 7. Page 5, line 30-32. I read this section a number of times before I understood that there were a series of retrieval experiments performed, in which the SSA in one of the three snowpack layers was allowed to be spatially variable in the truth. You might say that Figure 3 represents an example of the windslab layer being spatially variable in the truth, and that a-e represent the five histogram classes. 8. Page 5, line 30-31. I don't understand what this means. Please clarify exactly how the retrieval is performed. Is it essentially an iterative, Newton-Raphson type approach, that requires a first guess? And please clarify what "first guess" on line 31 means in this context. This sentence makes me think that for identical experiment parameters (i.e. for the same layer to be studied and same depth) you repeatedly changed the arbitrary first guess to the iterative algorithm, to see whether it is more or less independent from the first guess. I don't think this is what was done, however, based on the rest of the paper, so please clarify! 9. Page 6, line 8. "A perfect retrieval was assumed possible (negligible noise)". This may be confusing for readers, since the paper is based on diagnosing imperfect retrievals, and because there are many sources of noise. Do you just mean that you

assume perfect measurement of radar backscatter is possible, and thus you do not perturb the synthetic observations with white noise? 10. Page 6, line 14. Writing "CF(SWE)" implies that the cost function has one independent variable: snow water equivalent. At a mechanistic level, I don't think that squares with the paper. It seems to me that there are two inputs that are varied in the cost function in this study: depth, and SSA. I'm assuming that density is treated as known. 11. Page 6 line 16 "the estimated microstructure is a function of the SSA". So, a specified SSA value is passed to a function f(SSA), and then that is used to estimate exponential autocorrelation length? If that's correct, please state it. However, to keep this simple, I think you could note somewhere exactly how SSA is transformed to correlation length, and then when you write the cost function just have the input be SSA. 12. Page 8, line 30. "Notable differences . . . in different layers". Can you be more specific than "different layers"? I assume you're referring to the three layers assumed in the snowpack: depth hoar, wind slab, and surface snow. 13. I don't understand the equation in Table 1. If you plug in a depth of anything greater than ∼0.7 cm, you get a negative number and the surface snow percentage comes out as zero. Is depth intended to be in meters there? Additionally, can I recommend laying out the equation in the paper, and referencing it in the table? It's a little confusing with the way it's formatted in the table.

---

## Author Comment (AC1) · 19 Sep 2019

**"Effect of snow microstructure variability on Ku-band radar snow water equivalent retrievals" by Nick Rutter et al.**

We would like to thank the editor and the two anonymous referees for taking their time to read and comment on our original manuscript. We now provide a response to each of these comments. To help differentiate our responses, the text of the reviewer's comments are in black font and our responses are in blue.

Anonymous Referee #1

General comments: The authors present a well-written and well-reasoned study into the effects of snow microstructure on Ku-band SWE retrieval in sub-Arctic and Arctic landscapes.

Thank you for your overall positive judgement on this manuscript.

Please define the acronyms SWE and SSA at their first use in the Introduction.

Done.

Detailed comments

Figure 5: What are the "histograms of individual land surface types (red)"? Please clarify. These seem not to be discussed in the text either.

Figure 5 caption was previously: "Figure 5: Snow depths (limited to 2m) in TVC from airborne lidar: a) to c) histograms of individual land surface types (red) overlaid on the histogram of all snow depths (blue); d) distributions of snow depth by land surface type: blue box (inter-quartile range), red line (median), whiskers (dashed lines) extend from the end of each box to 1.5 times the interquartile range, outliers beyond this range are omitted."

Changed to: "Figure 5: Snow depths (limited to 2m) in TVC from airborne lidar: a) to c) histograms of three topographically delimited subdomains of the TVC catchment (red) overlaid on the histogram of all snow depths (blue); d) distributions of snow depth by land surface type: blue box (inter-quartile range), red line (median), whiskers (dashed lines) extend from the end of each box to 1.5 times the interquartile range, outliers beyond this range are omitted."

As topography (i.e. slope angle) has strong control on snow depth in tundra environments, linking the location of the trenches and pits to the distribution of different topographically delimited subdomains is important to illustrate the representative nature of the measurements across the whole TVC catchment, as well as the wider Arctic tundra environment. These are already discussed in the results section: "Figure 5 compares statistical distributions of snow depths from lidar in three topographically delimited subdomains (Figure 1: flat upper plateau, slope, flat valley bottom) of the TVC catchment. Median snow depths were very similar (0.60-0.65 m) across all subdomains. The inter-quartile range of snow depths on slopes was largest, reflecting enhanced drift processes that preferentially redistributed SWE to wind-sheltered areas of sloping terrain. However, similarity in frequencies of snow depths greater than 0.9 m between flat upper plateau and sloped subdomains suggested preferential SWE deposition also occurred over relatively flat (<5°) terrain, in addition to drifts in leeward slopes aligned perpendicular to prevailing north-westerly winds (King et al., 2018). This is of importance as flat upland plateau, where the majority of trenches and pits were located, was the areally dominant subdomain (66% of the total TVC lidar coverage in contrast to 19% slopes and 8% flat valley bottom). Consequently, field measurements well represented both the range and frequency of the total TVC lidar snow distribution. This type of exposed, flat, largely unforested

terrain is representative of pan-Arctic tundra environments, allowing potential for implications to be drawn beyond TVC."

P7, L11: What is a layer "entity"? Please define/describe. Their number is indicated in Table 1, suggesting that they are a significant element; however, they do not seem to be discussed in the text.

Layer entities are currently defined in the text as follows: "Thickness of tundra snowpack layers was spatially variable and frequently laterally discontinuous. Layers expand and contract horizontally, often creating several different discrete entities of the same layer across a trench face (Figure 6)."

The use of new expression ('entities') defined here allows the discontinuous nature of the layering in tundra snowpack to be expressed. For example, the 50 m trench (trench 4) will not have 36 layers as traditionally conceptualised in a 1-D snow pit profile. However, the horizontally discontinuous nature of the layers can be expressed by the number of coherent entities (36) of a smaller number of layers present in the tundra snowpack.

P9, L3: Reference to Figures 9x should be 10x. P9, L11: Should this be ". . .Figure 10 d and f). . ."? Also, should Figure 10b have a white line?

Figure cross-references have been corrected. There is no white line in Fig 10b as the retrievals are uniformly perfect for the range shown.

P11, L23: Although the use of single median values for density is reasonable, and density is not the focus of the paper, please comment on the relative effects that density may have on SWE retrieval error. That is, what would Figures 10b,d,f look like if density were to be perturbed within their interquartile ranges?

As long as the density is known the results are not materially different, hence this isn't included in the manuscript. However, the two figures below show mean density +/- one standard deviation. For the mean density plus one standard deviation, the SS / WS / DH densities are 195, 367, 303 kg m$^{-3}$ respectively, whereas for the mean density minus one standard deviation the densities are 53, 259, 209 kg m$^{-3}$.

[Figure]

Mean density +1 standard deviation.

[Figure]

Mean density -1 standard deviation.

In the Discussion, please also comment on the likely influence that ice lenses (such as the one mentioned on Page 6, Line 24 for 2017/18) may have on SWE retrieval error.

While there is plenty in the literature on the influence of ice lenses on passive microwave emission, there are very few published studies that include the impact of ice lenses on active microwave backscatter (e.g. Drinkwater et al., 2001: https://doi.org/10.1029/2001JD900107; Rott et al., 1993: https://doi.org/10.3189/S0260305500013070), and no data sets that we are aware of contain sufficient microstructural data to be able to evaluate the simulation of ice lenses in SMRT. Any impact of ice lenses on SWE retrieval will depend on the retrieval mitigation strategy and the focus of this paper is to demonstrate how the new dataset may be used rather than define a retrieval framework so it would be too speculative to comment on a likely influence on SWE retrieval. However, there is a clear need for field observations to test this and we have amended the following text in the methods section to reflect this: "While ice lenses were occasionally present in the 2018 snowpack and volumetric field samples of snow density and SSA contained sections of ice lenses, their impact on backscatter was not explicitly modelled by SMRT. There is a need for detailed field

measurement of ice lenses coincident with radar measurements as a priority for future SMRT evaluation. Ice lenses may be simulated in SMRT in terms of dielectric discontinuities, although coherent effects as a result of ice lens thickness are yet to be included.

Anonymous Referee #2

The authors present a really nice study of the effects of horizontal spatial variability of snow microstructure on retrieval errors. The study is in two parts: first a detailed description of snowpack properties measured in Trail Valley Creek, and second a synthetic retrieval experiment, where synthetic radar observations are generated using a radiative transfer model forced by realistic snow stratigraphy, and then those synthetic observations are processed with a retrieval algorithm. They show that SSA needs to be known quite precisely a priori in order to hit the target accuracy requirement.

The first part is wonderful: it shows in depth the spatial variability of observed properties, and occupies nearly all of the figures and tables. The second part I found hard to understand, so am asking for clarifications. To the extent I understand it, I think it is a critical contribution to this field!

In such studies, the details of the synthetic experiment design can make a lot of difference. After multiple re-reads, I had some trouble piecing together what exactly was done. All of my minor comments below are requests for clarification.

Thank you for your very positive judgement on the quality and scientific value of this manuscript.

My understanding is that this study performs a set of idealized depth retrieval experiments in order to isolate the impact of a single phenomenon (spatial variability of SSA in a single layer) on retrieval accuracy. It assumes (in my understanding) 1) perfect radar observations; 2) perfect knowledge of snow density; 3) perfect knowledge of background (i.e. soil and other substrate properties) 4) perfect ability to transform SSA into exponential autocorrelation length, and 5) perfect knowledge of SSA in two of the three layers in the snowpack. My understanding is that only depth is estimated by the retrieval, then transformed to SWE; SSA is assumed to be given. The study then systematically varies a spatially homogenous SSA value provided as a constant to the retrieval, and estimates depth, transforms to SWE, in order to compute the error metrics shown in Figure 10.

The reviewer is correct in their interpretation, and the following paragraph has been added to the end of the methodology section:

"In summary, the methodology was used to isolate the impact of the spatial variability of SSA in a single layer on retrieval accuracy from all other sources of retrieval error. Therefore, the following was assumed: 1) perfect radar observations; 2) perfect knowledge of snow layer densities; 3) perfect knowledge of the soil; 4) correct transformation of SSA into exponential autocorrelation length, and 5) perfect knowledge of SSA in all but one layer in the snowpack. Single (homogenous) values were used to represent the unknown SSA of the remaining layer. Only depth is estimated by the retrieval, which is then transformed to SWE via the known densities to compute the SWE retrieval error."

Anyway please clarify these minor points! I look forward to reading a revised version. Minor Comments

1. Page 5, line 11. Please clarify somewhere that density is assumed to be known, i.e. it is not being estimated by the retrieval algorithm, and you are giving the radar simulations for the "retrieval scene" the true density.

*We have now added the sentence: "Median observed layer densities used for the truth simulations were also used for the retrieval backscatter and SWE calculations."*

2. Page 5, line 11. Please clarify somewhere that SSA is treated as a specified input in the retrieval, if that is the case. I'm assuming that it is treated as "fixed" in the retrieval, in other words, you systematically specify a range of values, but the retrieval algorithm is not actually trying to estimate it. I'm also assuming that for each "layer" experiment, SSA in one layer is treated as spatially variable in the truth (using eqn 1), and is varied systematically in the retrieval scene (as shown in Figure 10), but that the other two layers are not only treated as spatially homogenous in the truth, but are also the "retrieval" simulations are given the true value of SSA. Please clarify this! I've read through a number of times but cannot find that information.

*We have now added the sentences: "For the spatially constant layers, the same SSA was used in both truth and retrieval simulations. For the layer with spatially varying SSA, a range of values were used to represent an unknown homogeneous SSA. As such, the retrieval does not attempt to retrieve SSA directly, but comparisons of the SWE retrieval errors for a given true snow depth allows selection of the SSA that best represents the spatially variable truth."*

3. Page 5, line 13-14. "Up to three layers were assumed within the snowpack". Can you reword this? I found it really confusing.

*This is really a function of the observed snow properties presented in the Results, but the following clarification has been added: "Three layers were assumed within the snowpack for snow up to 0.7m in depth: depth hoar (DH), wind slab (WS) and surface snow (SS), with layer thickness dependent on total snow depth based on relationships derived from snow trenches. For deeper snow, generally only wind slab and depth hoar layers were found (see section 3.1), so only two layers were simulated for snow depths greater than 0.7m."*

4. Page 5, line 16. "Horizontally homogenous snow was assumed for the retrieval". Please just clarify more explicitly here that you consider horizontal spatial variability in the truth, but horizontal spatial homogeneity in the retrieval.

*We have re-worded this as: "Whilst spatial variation in SSA was represented in one layer in the truth simulations, horizontally homogeneous snow was assumed for all layers in the retrieval, with snow depth retrieved as a function of the estimated snow microstructure."*

5. Page 5, line 20-21. Please say why you chose 5 intervals? Readers may assume this maps to "landforms" described in page 4, lines 27-29. I assume 5 is more or less arbitrary, or minimum needed to capture spatial observed distributions, which is fine, but please clarify. Also, this is a great chance to explicitly say that the set of 5 simulations represent spatial variability in that the frequency and weights represent the proportion of a scene that might take each SSA value.

*We have added the following sentence to improve clarity: "Five intervals were chosen to describe the SSA distribution adequately whilst maintaining simplicity."*

*and*

*"Aggregated together, the set of 5 simulations represent spatial variability in that the frequency and weights represent the proportion of a scene that might take each SSA value."*

6. Page 5, line 21. "across the observed range". This is referring to the in situ datasets of SSA, correct? But readers could easily get confused as this is how you are computing the synthetic radar observation. Please clarify the language?

Changed to: "across the observed SSA range".

7. Page 5, line 30-32. I read this section a number of times before I understood that there were a series of retrieval experiments performed, in which the SSA in one of the three snowpack layers was allowed to be spatially variable in the truth. You might say that Figure 3 represents an example of the windslab layer being spatially variable in the truth, and that a-e represent the five histogram classes.

The following sentence has been added: "A set of three experiments were performed that considered spatial variability in SSA in each of the DH, WS and SS layers in turn."

8. Page 5, line 30-31. I don't understand what this means. Please clarify exactly how the retrieval is performed. Is it essentially an iterative, Newton-Raphson type approach, that requires a first guess? And please clarify what "first guess" on line 31 means in this context. This sentence makes me think that for identical experiment parameters (i.e. for the same layer to be studied and same depth) you repeatedly changed the arbitrary first guess to the iterative algorithm, to see whether it is more or less independent from the first guess. I don't think this is what was done, however, based on the rest of the paper, so please clarify!

The reviewer has perfectly captured our internal discussion of use of jargon 'a priori' vs potentially misleading 'first guess'! Please see response to reviewer #2 question 2. This has been rewritten as 'unknown homogeneous SSA', and now referred to as 'unknown snow microstructural characteristics' elsewhere in the document.

9. Page 6, line 8. "A perfect retrieval was assumed possible (negligible noise)". This may be confusing for readers, since the paper is based on diagnosing imperfect retrievals, and because there are many sources of noise. Do you just mean that you assume perfect measurement of radar backscatter is possible, and thus you do not perturb the synthetic observations with white noise?

We have changed this to be more explicit: "observations were assumed (i.e. no noise added to truth backscatter)".

10. Page 6, line 14. Writing "CF(SWE)" implies that the cost function has one independent variable: snow water equivalent. At a mechanistic level, I don't think that squares with the paper. It seems to me that there are two inputs that are varied in the cost function in this study: depth, and SSA. I'm assuming that density is treated as known.

Equation 3 has been rewritten to use CF(d, SSA)

11. Page 6 line 16 "the estimated microstructure is a function of the SSA". So, a specified SSA value is passed to a function f(SSA), and then that is used to estimate exponential autocorrelation length? If that's correct, please state it. However, to keep this simple, I think you could note somewhere exactly how SSA is transformed to correlation length, and then when you write the cost function just have the input be SSA.

The following sentence has been added: "For these simulations the exponential autocorrelation length used in the SMRT simulations (both truth and retrieval) is calculated with equations 5 and 10 of Mätzler (2002)."

12. Page 8, line 30. "Notable differences . . . in different layers". Can you be more specific than "different layers"? I assume you're referring to the three layers assumed in the snowpack: depth hoar, wind slab, and surface snow.

Previously: "Notable differences between distributions of SSA and density in different snowpack layers allowed parametrisation of snowpack microstructure in the SMRT model"

Changed to: "Notable differences between distributions of SSA and density in surface snow, wind slab and depth hoar layers allowed parametrisation of snowpack microstructure in the SMRT model"

13. I don't understand the equation in Table 1. If you plug in a depth of anything greater than ~0.7 cm, you get a negative number and the surface snow percentage comes out as zero. Is depth intended to be in meters there? Additionally, can I recommend laying out the equation in the paper, and referencing it in the table? It's a little confusing with the way it's formatted in the table.

Thank you for spotting this. The equation in the table has been reformatted to:

$$\Delta z^{SS} = \begin{cases} -44.7269\, d + 30.1551, & d < 0.7 \\ 0, & d \geq 0.7 \end{cases}$$